# Mobile Application Based Pelvic Floor Muscle Training for Treatment of Stress Urinary Incontinence: An Assessor-Blind, Randomized, Controlled Trial

**DOI:** 10.3390/jcm12227003

**Published:** 2023-11-09

**Authors:** Athasit Kijmanawat, Apisith Saraluck, Jittima Manonai, Rujira Wattanayingcharoenchai, Komkrit Aimjirakul, Orawee Chinthakanan

**Affiliations:** Department of Obstetrics & Gynaecology, Faculty of Medicine, Ramathibodi Hospital, Mahidol University, Bangkok 10400, Thailand; athasit.kij@mahidol.ac.th (A.K.); apisith.sar@mahidol.ac.th (A.S.); jittima.man@mahidol.ac.th (J.M.); rujira.wat@mahidol.ac.th (R.W.); komkrit.aim@mahidol.ac.th (K.A.)

**Keywords:** mobile apps, stress urinary incontinence, pelvic floor muscle training, randomized controlled trial

## Abstract

A first-line treatment for stress urinary incontinence (SUI) is pelvic floor muscle training (PFMT) for at least three months. The key problem is that patients do not understand the importance of these exercises and their effectiveness. Mobile health apps offer new possibilities to increase treatment adherence. This study compared a reduction in SUI, exercise adherence, and quality of life in PFMT with animation vs. standard instruction. A prospective, single-blind, randomized control trial was collected. SUI patients were randomized into the application or control groups confirmed using a one-hour pad test. In the intervention group, the PFMT application was applied via mobile phone (PFMT with animations, recording system, and reminder system). The standard exercise protocol was similar in both groups. Additional follow-up was conducted at 4, 8, and 12 weeks. A total of 51 participants were randomized to the application (n = 26) and control groups (n = 25), respectively. At the 12-week follow-up, there was no significant difference between the two groups in terms of SUI cure rate, SUI severity by pad test, and daily SUI episodes from the bladder diary (*p*-value of 0.695, 0.472, and 0.338, respectively). The mean PFME adherence in the application group was higher than the control group at 8 weeks (66.3 ± 13.6 vs. 52.7 ± 16.6, *p* = 0.002) and 12 weeks (59.1 ± 13.9 vs. 37.8 ± 11.0, *p* = 0.001). The application group reported no difference from the conventional PFMT group in terms of improvements in SUI cure rate, symptom severity, and quality of life effects at 12-week follow-up. However, the improvement evaluated by the mean difference in SUI episodes and quality of life effects (ICIQ-UI SF) reported a better outcome in the mobile app group. The PFMT application has been proven to be an effective tool that improves PFMT adherence.

## 1. Introduction

Urinary incontinence (UI) is a prevalent issue among women, which has a significant impact on overall health, quality of life, and mental well-being [1,2]. Stress urinary incontinence (SUI) is the prevailing disorder in urinary storage [2]. Stress urinary incontinence (SUI) in women, characterized by the involuntary leaking of urine during physical exertion, is the prevalent type of incontinence among women, accounting for approximately 86% of reported cases [3]. SUI is a very common problem and can be found in women of all age groups [2]. It occurs when increasing intraabdominal pressure, such as coughing, sneezing, laughing, exercising, lifting heavy objects, and doing other physical movements. Its causes include conditions that result in physical changes, such as pregnancy and menopause [4]. A significant proportion of women continue to possess limited understanding regarding stress urinary incontinence (SUI) and the availability of successful alternatives to treatment [5]. Wide disparities in prevalence rates for female SUI have been reported. The prevalence was found to be 35% in European countries [6], 23% in Spain, and 44%, 41%, and 42% in France, Germany, and the United Kingdom, respectively. Interestingly, 56% of Asian women were reported to have female SUI [7]. The reported prevalence of female SUI in Thailand ranged from 20 to 41% [8]. SUI in women is one of the most significant issues affecting daily life and quality of life in terms of physical and mental health [9,10]. The literature reports negative effects on psychosocial well-being, mental health, and health-related quality of life [11]. In addition, female SUI may trigger women to experience anxiety and depression due to social embarrassment [12]. In addition, female SUI has detrimental effects on sexual function [13,14].

According to the guidelines developed by the International Consultation on Incontinence (ICS) on the treatment of stress urinary incontinence, pelvic floor muscle training (PFMT) needs to be the initial standard treatment for non-surgical treatment [15]. The strengthening of the pelvic floor muscles is a traditional part of the conservative therapy for female stress urinary incontinence. It is a contraction of the pelvic floor muscles that is performed intentionally and skillfully. It entails pulling the pelvic floor muscle up and forward before and during an effort or exertion in order to increase its effectiveness. [16] In addition, having toned and robust pelvic floor muscles makes the neck of the bladder more resistant to straining, which provides support for the area. The meta-analysis and systematic review both revealed that PFMT, regardless of the protocol that was utilized, led to a reduction in the amount of urine lost in women who suffered from SUI. On the other hand, for the program to have a significant impact, it ought to continue for 6–12 weeks; there should be more than three sessions each week, and each session should last for no more than 45 min [17]. On the other hand, the objective of pelvic floor muscle training is to strengthen the muscle through contraction and relaxation of the pelvic floor muscles. In some recommendations, pelvic floor muscle exercises were suggested to be performed 2–4 times per day, 8–12 times per day, or at least 2–3 times per week for a minimum of three consecutive months [18,19].

Even though PFME is an effective and non-invasive treatment, adherence is an important issue. Behavioral change, such as adherence, is an extraordinarily complicated procedure, and sections of it have been conceptualized in various theoretical models of behavior [20]. Consequently, clinicians were more likely to utilize theoretical frameworks such as social cognitive theory, stages of change, or the trans-theoretical model. The integrated model of behavior change implies three aspects in the process of change: awareness, motivation, and action [21]. For this reason, pelvic floor muscle training could be effective in changing behavior in women with stress urinary incontinence and a high rate of adherence who have awareness, motivation, and regularity of action. In general, the level of commitment to pelvic floor muscle exercises tends to decrease with time. A prior investigation revealed that the conventional approach to instruction just inside the hospital setting had a suboptimal level of adherence to pelvic floor muscle exercises (PFME) [22]. Women continued exercising well early in the treatment and gradually decreased over time to less than 50% at 3 months [23]. One of the primary issues is patients’ lack of comprehension of the impact of exercise, leading them to see it as less efficacious. Consequently, this perception results in inconsistent engagement in pelvic floor muscle exercises [24].

By using animated materials for annotations, they can be more understandable than just descriptions, which affects the key behavior modification to regular exercise, which comprises two factors: exercise motivation, which is caused by building a knowledge and understanding of the principles, methods, and benefits of exercise, as well as correcting misconceptions and setting goals for exercise [25]. The second is constant exercise. It was found that regular stimulation and self-evaluation led to more consistent exercise [26]. A consensus demonstrated that it was important to investigate the use of technology, such as electronic-based resources, to facilitate and maximize PFMT compliance [27]. A strategy to remind women to use a mobile application is likely to be effective. This study used a mobile device application (app) to evaluate the use of technology in urinary incontinence in a randomized controlled trial. Questionnaires revealed significant improvements in SUI. In addition, the authors discovered an increase in app group adherence [28]. One study analyzed the factors associated with the effectiveness of an app and found that women who are more interested in the treatment and have high expectations for it have better outcomes [19]. In contrast, a previous study evaluated an Internet-based treatment and concluded that a lack of face-to-face contact was not a barrier to adherence among women using this technological strategy [29]. In addition, an observational study of women with UI who use mobile apps revealed that 65.6% of them had UI improvement. [30]. In terms of less severe incontinence, the frequency of PFMT and the use of an app were associated with significant improvement. However, this study was not compared with the standard procedure. In Sweden, another study found that a new, cost-effective first-line treatment for SUI using a mobile app could make it easier for this group of patients to attain care in the long term [31]. An American Journal of Obstetrics and Gynecology article published in 2022 stated that mobile applications provide varying degrees of content, function, and overall utility, as well as limited disease information and decision support. [32]. None of the apps had all of the desired features, so it is essential to study the efficacy of mobile apps on female SUI in order to gain a better understanding of their effects and limitations as a tool for pelvic floor muscle training. Therefore, the authors expected that the pelvic floor muscle exercise teaching, aided by using an animation program that includes teaching systems, self-assessment, and motivation to exercise, would make it possible to alleviate stress urinary incontinence symptoms better than the standard teaching method. This study aims to compare the effects of the mobile pelvic floor muscle exercise application and the conventional method (control group) on the cure rate of female SUI as measured by objective outcome (one-hour pad weight test) and subjective outcome (the frequency of SUI episodes, ICIQ-IU SF scores, PGI-I scores), as well as patients’ adherence to pelvic floor exercises at home over a 12-week period.

## 2. Materials and Methods

A prospective, randomized, control trial, single-blind study was conducted at the Female Pelvic Medicine and Reconstructive Surgery Unit, Department of Obstetrics and Gynaecology, Ramathibodi Hospital, from September 2020 to April 2021. This study was reviewed and approved by the Committee on Human Rights Related to Research Involving Human Subjects, Faculty of Medicine, Ramathibodi Hospital, and complied with the Declaration of Helsinki and Thai clinical trials registry TCTR20230802004. Written informed consent was obtained from all participants.

Participants were recruited through hospital recruitment from the Female Pelvic Medicine and Reconstructive Surgery Clinic. Potential participants presenting with SUI symptoms were requested to meet with a research assistant, who provided them with comprehensive study information and an enrollment form. The inclusion criteria included: (i) SUI confirmed using a one-hour pad test of at least 1 g (ii) age ≥ 18 years old; (iii) not pregnant or postpartum period (iv) pelvic organ prolapse less than stage II by POP-Q examination (v) no history of chronic diseases, such as kidney disease or neurological disease (vi) be able to communicate in Thai and use the mobile application. Exclusion criteria included inability to contract the pelvic floor muscles, urinary tract infection, and urgency urinary incontinence. After providing verbal informed consent to participate in the study, participants completed screening questionnaires and reported the number of SUI events and the number of napkins used daily. Subsequently, they were given a pelvic exam, POP-Q, stress test, 1-h pad-weight test (PWT), post-void residual urine, and urine analysis. An assessment of pelvic floor muscle strength using the Brink scale was also conducted. At last, the principal investigator verified that all data strictly met the inclusion criteria. The diagnosis of SUI was established based on the presence of clinical symptoms, specifically urine leakage during physical effort or activity, in the absence of urgent symptoms or urge with a positive one-hour pad weight test. A one-hour pad test was performed by instructing a patient to drink 500 mL of water within 15 min, then the patient was put on the urine pad and performed the following activities: walk for about 30 min up and down the stairs, 10 stand-up and sit down exercises, 10 jumps, 10 continuous coughs, wash your hands with the sound of running water. Then, the extra weight in the urine pad was measured. If the weight increases by 1 g, it is considered positive. Women who met the eligibility criteria were invited to participate in this study and signed the written informed consent. Additionally, data on using validated the International Consultation on Incontinence Questionnaire-Urinary Incontinence Short Form (ICIQ-UI SF) score [33] were gathered from the included patients. Baseline demographic and clinical data of the participants, including age, BMI, parity, vaginal examination, and pelvic organ prolapse, were collected according to a participant data form. Participants were randomly assigned into either the mobile application or home PFMT groups using a block of 4 randomizations to avoid bias from the sequence of the procedures (Figure 1). The allocation sequence was generated by a statistician who was not involved in this study using a computer-generated random number table in a ratio of 1:1 and was concealed in sealed opaque envelopes that were sequentially numbered. The envelopes were stored in a closed locker in the clinic to which only the research assistant had access. The envelopes were given to the principal investigator immediately at the time of recruitment.

### 2.1. Sample Size Estimation

The determination of sample size was predicated upon the primary outcome, which is the rate of SUI cure as measured using a one-hour pad test. Based on the findings of Fitz FF et al. [22], the SUI cure rate in standard care was 28%. A difference in SUI cure rate between the two interventions (mobile app vs. home PFMT) was set at 40%. Given a statistical power of 80%, a two-sided significance threshold of 0.05, and a ratio of 1:1, the calculated sample size was 48 (24 per group). After taking into account a 20% data loss, the overall sample size was determined to be 60, with 30 participants allocated to each group.

### 2.2. Application Group

In the application group, a pelvic floor muscle exercise application was downloaded via mobile phone. Participants were taught to perform pelvic floor exercises using animations and instructed on how to use the application. (Figure 2) The pelvic floor muscle exercise application included (i) an animation of pelvic floor anatomy, causes of SUI, mechanisms of pelvic floor exercises in reducing SUI, and how to exercise your pelvic floor muscles, (ii) a vibration and sound alert system for participants to exercise their pelvic floor muscles for a specified period of time. (exercise every day, 3 rounds a day, 5–10 min per round), (iii) a pelvic floor exercise recording system which was able to help them compete with other participants who were using the same application. To observe adherence in the application group, the researcher accessed the application records to determine how often the protocol program was activated.

All participants received standard brochures explaining the method of pelvic floor exercise and noting the exercise frequency. The exercise protocol was the same in the application and control groups. The completed protocol was defined as performing exercise every day, 3 rounds a day, 5–10 min per round (sitting, lying down, or standing) for 12 weeks. The incontinence specialist nurse taught PFME using the same procedures and distributed the same questionnaire to all participants. Additional follow-up was conducted at 4, 8, and 12 weeks.

### 2.3. Control Group

Participants in this group received printed instructions for home PFMT. They were asked to fill in a diary offering information about adherence during home exercise.

### 2.4. Main Outcome Measures

The primary outcome was to compare the SUI cure rate defined by a negative one-hour pad test (less than 1 g of urinary leakage) between the application group (pelvic floor muscle exercise instruction with mobile application) and the control group (pelvic floor muscle exercise instruction). Furthermore, the reduction in SUI, assessed using a one-hour pad test, was also compared.

The secondary outcomes were to compare SUI episodes by bladder diary, patients’ satisfaction, and patients’ quality of life effects and were evaluated using the ICIQ-UI SF score [33] and the Patient Global Impression of Improvement (PGI-I) [34], respectively.

The evaluation of adherence to pelvic floor muscle exercise was evaluated. The adherence to PFME was calculated from the formula (number of contractions performed/expected number of contractions) × 100%. Then, it was divided into groups as follows: high (>80%), moderate (20–80%), and low (<20%) adherence. The expected number of contractions is defined as daily exercise, 3 rounds a day, and 5–10 min per round. The strength of the pelvic floor muscles was assessed using the Brink scale.

Statistical analysis was performed using PASW Statistics version 18.0 (SPSS, Inc., Chicago, IL, USA). Quantitative variables, including age, BMI, parity, SUI episodes, POP-Q stage, one-hour pad test, and ICIQ-UI SF score. The Brink scale and adherence percentage were expressed as mean ± standard deviation (SD) or median and interquartile range. Qualitative variables, including SUI cure rate, menopausal status, and history of POP surgery, were expressed as frequency and percentage. For quantitative variables, a comparison between intervention groups was performed by an independent *t*-test or Mann–Whitney U test, where appropriate. The chi-square test was used to compare the groups in relation to the qualitative variables. A *p*-value of less than 0.05 was considered statistically significant.

## 3. Results

One hundred and forty-two women with SUI were screened; 67 women did not meet the inclusion criteria, and 15 declined to participate. This study included 60 women who were randomized into either the application group (n = 30) or the control group (n = 30). One participant from the application group and two from the control group were lost to follow-up at 8 weeks after intervention. Three participants from each group were lost to follow-up at 12 weeks. Fifty-one women (the application group, n = 26; the control group, n = 25) completed 12 weeks of the study. Figure 1 shows the flow diagram for study participants. There were four (6.7%), 43 (71.7%), and 13 (21.6%) participants who had been suffering from mild, moderate, and severe SUI, respectively. All women assigned to the application group who completed the follow-up (n = 26) had used the app. The animation for pelvic floor muscle exercise, reminder setting, and pelvic floor muscle exercise (PFME) record was activated at a 100% rate of use.

The mean (SD) age of participants in the application group was 53.0 (9.3) years, whereas in the control group it was 56.5 (8.9) years (*p* > 0.05). The majority of participants indicated parous status, with a median parity of 2. The body mass index (BMI) values (mean, SD) for both the mobile app group and the control group were 24.8 (3.5) kg/m^2^ and 24.7 (3.8) kg/m^2^, respectively. The difference in BMI between the two groups was not statistically significant (*p* = 0.941). The mean scores (SD) on the Brink scale, which assesses pelvic floor muscle strength, were found to be similar between the two groups. Specifically, the mean score was 6.5 (1.5) in the group using mobile apps and 6.9 (1.4) in the control group. These scores indicate that all participants had an adequate degree of pelvic floor muscle tone, which is a crucial factor for the intervention being studied. The mobile app group exhibited a median (IQR) of 1.9 (1.0, 3.3) for reporting SUI events, whereas the control group displayed a median of 1.5 (0.7, 3.7). The median values of the pad weight test before intervention in the mobile app and control groups were 4.6 (IQR 3.0 to 8.1) and 4.3 (IQR2.7 to 10.0), respectively. The mean (SD) initial scores of the ICIQ-UI SF were 8.5 (2.6) and 8.0 (2.3) in the mobile app group and control group, respectively. Considering SUI symptoms, the number of incontinent episodes per day, pad test, and the ICIQ-UI SF score of the application and the control groups were not significantly different before the intervention (*p* > 0.05) (Table 1).

The primary outcome of this study was presented by an objective assessment utilizing the one-hour pad weight test. The proportion of individuals with a negative result in the pad weight test after intervention (less than 1 g) was 30.8% (8 out of 26) in the application group and 36% (9 out of 25) in the control group. (*p* = 0.695). The median (IQR) of the mobile applications group decreased from 4.6 (3.0, 8.1) to 3.2 (0.1, 6.5), whereas the median (IQR) of the control group decreased from 4.3 (2.7, 10.0) to 3.5 (0.6, 8.3). When comparing the mean difference before and after treatment between the mobile apps group and the control group, the mean differences (SD) were −1.8 (−2.9, −0.4) and −1.6 (−3.0, 0), respectively. Regarding the main endpoints following a 12-week treatment period, there was no statistically significant difference seen between the two groups in terms of the median weight for the one-hour pad test (*p* = 0.727) (Table 2).

Secondary outcomes were assessed by evaluating subjective symptom reports from participants. The median (IQR) of SUI episodes before and after treatment in the mobile applications group decreased from 1.9 (1.0, 3.3) to 0.9 (0, 1.7), whereas in the control group, it decreased from 1.5 (0.7, 3.7) to 1.0 (0, 3.0). It is interesting that upon analyzing the mean difference between the two groups, it was seen that the group utilizing mobile applications had superior reported results among participants in comparison to the control group, with statistical significance. The application group demonstrated a significant decrease in the negative impact of stress urinary incontinence (SUI) on quality of life, as measured by ICIQ-UI SF, compared with the control group. The median score (IQR) for the application group was −4.0 (−4.0, −2.0), while the control group had a median score of −2.0 (−5.5, −1.0) at week 12. This difference was statistically significant (*p*= 0.045). Regarding the patients’ satisfaction with both interventions, as determined by the PGI ratings. The average PGI-I score of the application group was 4.7 ± 1.9, which demonstrated a statistically significant increase compared with the control group (3.6 ± 1.9) (*p* = 0.004).

Table 3 and Figure 3 demonstrate adherence to PFME of both groups at weeks 4, 8, and 12. The mean PFME adherence in the application group was markedly higher than the control group at week 8 (66.3 ± 13.6 vs 52.7 ± 16.6; *p* = 0.002) and week 12 (59.1 ± 13.9 vs 37.8 ± 11.0; *p* = 0.001). More than half of the participants in the application group had been performing PFME regularly.

## 4. Discussion

In the contemporary era of digital transformation, the integration of technology into everyday life has emerged as a pivotal motivator for altering human behavior. The utilization of mobile applications has been utilized for several diseases to enhance and enhance the quality of therapy in conditions requiring behavioral change, such as patients with morbid obesity [35]. The initial therapy for female SUI is primarily focused on pelvic floor muscle training (PFMT), as recommended by worldwide guidelines [15]. The effectiveness of PFMT in achieving beneficial results has been demonstrated. Assessing the methods utilized to give support and encouragement to female SUI patients engaging in PFMT is an essential task for urogynecologists and healthcare providers. This initiative aims to enhance the standard of treatment and implement a comprehensive strategy towards patients in the current era. The development and utilization of mobile applications for patients with pelvic floor disorder is an innovative intervention that has garnered increasing interest worldwide. This intervention is believed to facilitate behavioral awareness and promote changes via the use of technological tools.

This study demonstrated the efficacy of a specialized pelvic floor muscle exercise application designed specifically for the treatment of stress urine incontinence. Although the utilization of a mobile application for pelvic floor muscle training (PFMT) in the present research did not yield statistically significant results in terms of objective measurements, as indicated by the one-hour pad test, participants in the group utilizing the application reported experiencing improvements in subjective measurements. They reported a reduction in symptoms, an enhanced quality of life, and a decrease in the frequency of leakage episodes per day. This study reveals a noteworthy finding on the beneficial function of mobile apps, specifically in terms of promoting exercise adherence. It is seen that individuals utilizing mobile apps exhibit higher levels of adherence to exercise regimens compared with those in the control group. During this study, the application was found to be user-friendly without major technical problems, and also no changes to the application. The majority of this study population had mild to moderate SUI, thus representing a clinically relevant group of women with incontinence. Most outcome measures were validated and highly recommended, which enabled comparisons with other studies. The loss to follow-up rate was low, with few missing outcome values, reducing the risk of bias that may arise when setting values.

Our results confirm data from a scientific study showing that PFMT improves symptoms of stress urinary incontinence [27]. Both groups achieved this goal. Meanwhile, studies have shown that home treatment is less effective than supervised treatment [36]. The authors highlight the importance of information, knowledge about PFM functions, and correct muscular contraction to achieve a cure by pelvic floor animation. The same finding was reinforced in our study in both groups, and it is possibly the reason why almost all of the participants demonstrated improvement in their symptoms. One advantage of the application aiding treatment is the reminder function, which may increase adherence to PFMT. In a study on adherence to PFMT, the most common barrier was difficulty in remembering to perform the exercises. The pelvic floor muscle exercise application effectively increases the adherence of women to perform pelvic floor muscle exercise, compared with the standard method, in the same way, that our study found that PFMT alleviates symptoms of urinary incontinence, according to the validated ICIQ-UI SF and the PGI-I questionnaires. Before treatment, participants in the app group had a mean ICIQ-UI SF score of 8.5 (SD 2.6); after treatment, the mean score was reduced by 3.9. This reduction was of the same magnitude as reductions previously reported. For example, two RCTs that tested different PFMT programs included participants with baseline ICIQ-UI SF scores of 8.6–12.0. After intervention applied, mean scores were reduced by 3.0–4.5 [37,38]. In those studies, participants ages (32–72) were similar to those in our study. Previously, the minimum significant difference established for conservative treatment for ICIQ-UI SF was 2.5 [18], so improvements above these levels were clinically relevant. In 2021, a similar RCT study on the use of a mobile app for female SUI will also report results. After two years, the mean decreases in ICIQ-UI SF and ICIQ-LUTS quality of life scores were 3.1% (interval of confidence: 2.0–4.2) and 4.0% (interval of confidence: 2.1–5.9), respectively. In terms of outcome measurement for SUI improvement and quality of life effects, the study’s findings are comparable to this one [39]. Another study utilizing app-based self-management for the treatment of urgency and mixed urinary incontinence in women was conducted using the Tat^®^II app. This study found that the mean ICIQ-UI SF score decreased from 11.5 to 7.6 (mean difference = 4.0, 95% CI = 3.2–4.7). Self-management with the UUI and MUI had a significant effect on all outcome measures and could serve as an alternative first-line treatment for these conditions, the researchers concluded. Even if these data reflected long-term follow-up, the single-arm study cannot describe how the conventional technique differs [40]. Interestingly, a study on the factors associated with successful treatment with a mobile app found that three factors were significantly associated with success: higher expectations for treatment, weight control, and self-reported improvement of pelvic floor muscle strength. Even though our study was not focused on risk factor analysis, it is essential to understand and comprehend the patients in order to set a common goal while using a mobile app [19].

PFMT adherence is complex and necessitates a change in behavioral motivation and the active involvement of women. The difference between short and long-term adherence to PFMT in previous studies is that clinicians estimate that 64% of patients adhere to short-term PFMT and health advice, but only 23% adhere to long-term exercise [5]. The success of PFMT in the long term (defined as 6 months of training) varies between 41% and 85% [25]. Two studies showed that only 15% to 28% of women continue PFMT after 10 to 15 years. Additional methods of PFMT can increase adherence and are useful in maintaining exercise [18]. The application for SUI treatment could provide benefit by a reduction in in-person visits for PFMT. Additionally, exercise adherence in the control group is lower than in the application group, while the application group can sustain performing PFMT regularly for a long time and decrease the need for additional treatments.

The strengths of our study were the randomized control trial design, assessor-blinded, comparison with standard treatment, use of standard outcome measurements, and two validated questionnaires. Furthermore, this study has a wide scope in measuring outcomes, encompassing both objective and subjective measurements. These obtained data can provide an extensive understanding of the impacts of the intervention on diseases and the overall care provided to patients. According to that, this study assessed the effectiveness of a mobile application for treating female SUI, which enhances patient behavioral modifications and increases adherence. The most notable strength of this study is the motivation to use the tools in something that humans are accustomed to using on a daily basis, such as a mobile phone. It is evident that this will be beneficial for implementation and use in a variety of settings and diseases. Our study has some limitations. We used one specific PFMT protocol, and individual differences were not considered. This study did not compare the application with another active treatment. Furthermore, only a small sample was assessed, and the follow-up was short, with some participants lost to follow-up. Following the analysis of our study findings, a power calculation was conducted utilizing two independent population proportions. The resulting power of this study was determined to be 5.7%. In order to enhance the statistical significance of the findings, it is recommended that the sample size for each group in this study be increased to a minimum of 1397 individuals due to the fact that this study marked the beginning of a trial using a mobile app to treat female SUI in Thailand, a trial that has rarely been conducted in the Thai population before. In this study, we have shown that application treatment is effective in the short term. Evaluations of the long-term adherence and effects are ongoing, as are studies on the cost-effectiveness of the application. Since this study showed that adherence decreased more in the control group than in the app group, a long-term follow-up study will be necessary to show the impact of adherence on urinary symptoms and SUI cure rate. This study’s findings have important clinical implications, suggesting that the use of mobile applications for the treatment of female SUI yields superior outcomes. Therefore, it is recommended to promote and implement the utilization of mobile apps for pelvic floor muscle training (PFMT) in female SUI patients, particularly those who can afford and access mobile applications with satisfactory performance. Based on the findings of this study, it is evident that there is a convincing case for further investigation into the development of medical engineering applications, techniques, and equipment aimed at enhancing the efficacy of PFMT for female SUI and a variety of pelvic floor diseases. This research study provides an incentive for future endeavors in this field. Furthermore, further research is recommended on the efficacy of various digital technologies, including animation, artificial intelligence, and devices.

Although technology cannot be a substitute for face-to-face human contact, the correct evaluation of pelvic floor muscle strength and tone, orientation on correct pelvic floor muscle exercise, and motivational reinforcement are essential for treatment success. There is no gold standard method for PFMT, and it varies in different settings. Also, we wanted to provide treatment accessibility to women. Pelvic floor muscle exercise application could be a complementary treatment tool. The application is suitable for women who want to try an easily accessible self-management treatment and reminder function.

## 5. Conclusions

The application group reported no difference from the conventional PFMT group in terms of improvements in SUI cure rate, symptom severity, and quality of life effects at 12-week follow-up. However, the improvement evaluated by the mean difference in SUI episodes and quality of life effects (ICIQ-UI SF) reported a better outcome in the mobile app group. The PFMT application has been proven to be an effective tool that improves PFMT adherence.

## Figures and Tables

**Figure 1 jcm-12-07003-f001:**
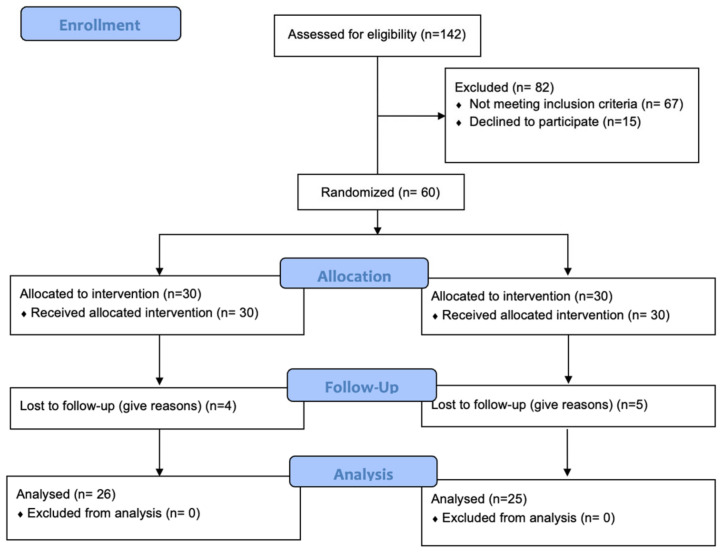
CONSORT flowchart of patients’ recruitment and study flow.

**Figure 2 jcm-12-07003-f002:**
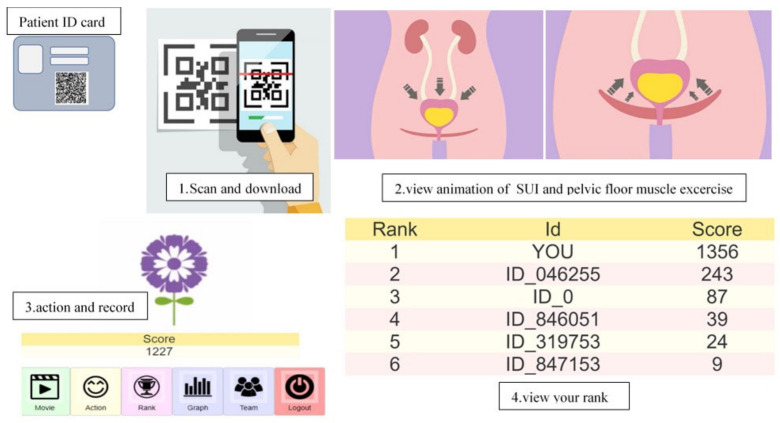
The pelvic floor exercise mobile application.

**Figure 3 jcm-12-07003-f003:**
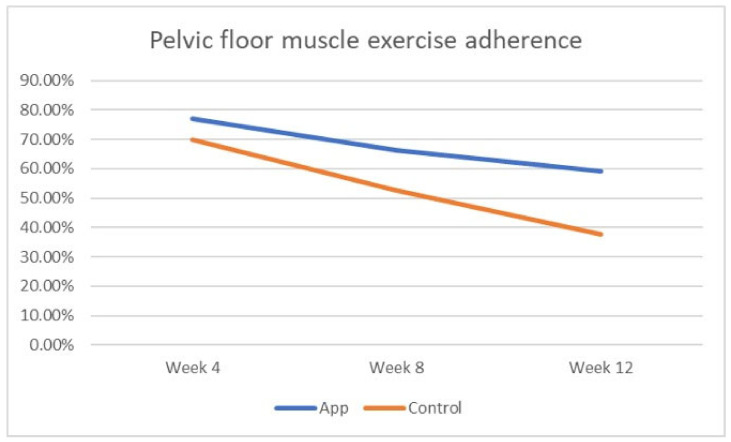
Pelvic floor muscle exercise adherence measures during the PFMT.

**Table 1 jcm-12-07003-t001:** Characteristics of treatment and control groups at pre-treatment.

Variable	App Group (n = 26)	Control Group (n = 25)	*p*-Value
Age (years), mean (SD) ^a^	53.0 (9.3)	56.5 (8.9)	0.139
BMI (kg/m^2^), mean (SD) ^a^	24.8 (3.5)	24.7 (3.8)	0.941
Parity, median (IQR) ^b^	2 (0, 3)	2 (1, 2)	0.731
POP stage median (IQR) ^b^ anterior compartment posterior compartment apical compartment	1 (1, 2) 1 (1, 1) 1 (1, 1)	1 (0, 2) 1 (0, 1) 1 (1, 1)	0.612 0.812 0.325
Pad test (grams), median (IQR) ^b^	4.6 (3.0, 8.1)	4.3 (2.7, 10.0)	0.982
SUI episode, median (IQR) ^b^	1.9 (1.0, 3.3)	1.5 (0.7, 3.7)	0.988
ICIQ-UI SF, mean (SD) ^b^	8.5 (2.6)	8.0 (2.3)	0.430
Brink scale, mean (SD) ^a^	6.5 (1.5)	6.9 (1.4)	0.298

^a^ Independent *t*-test analysis. ^b^ Mann–Whitney U test analysis.

**Table 2 jcm-12-07003-t002:** Pad test, SUI episodes, SUI cure rate, Brink scale, PGI-I, and ICIQ-UI SF score at week 12.

Variable	12 Weeks Follow-up		*p*-Value	Mean Difference		*p*-Value
	App Group n = 26	Control Group n = 25		App Group n = 26	Control Group n = 25	
SUI cure rate, n (%) ^c^	8 (30.8)	9 (36.0)	0.695	-	-	-
Pad test (grams), median (IQR) ^b^	3.2 (0.1, 6.5)	3.5 (0.6, 8.3)	0.472	−1.8 (−2.9, −0.4)	−1.6 (−3.0, 0)	0.727
SUI episode (times), median (IQR) ^b^	0.9 (0, 1.7)	1.0 (0, 3.0)	0.338	−0.9 (−1.6, −0.7)	−0.4 (−0.9, 0.2)	0.009
ICIQ-UI SF, median (IQR) ^b^	4.5 (1.0, 6.0)	5.0 (2.0, 7.5)	0.314	−4.0 (−4.0, −2.0)	−2.0 (−5.5, −1.0)	0.045
PGI-I score, mean (SD) ^a^	4.7 (1.9)	3.6 (1.9)	0.043	-	-	-
Brink scale, mean (SD) ^a^	8.5 (1.8)	7.2 (1.5)	0.007	2.1 (1.6)	0.3 (1.4)	0.433

^a^ Independent *t*-test analysis. ^b^ Mann–Whitney U test analysis. ^c^ Chi-square test.

**Table 3 jcm-12-07003-t003:** Adherence to PFME at weeks 4, 8, and 12.

Variable	Week 4		*p*-Value	Week 8		*p*-Value	Week 12		*p*-Value
	App Group n = 30	Control Group n = 30		App Group n = 29	Control Group n = 28		App Group n = 26	Control Group n = 25	
Adherence percentage, % (SD) ^a^	77.1 (14.0)	70.0 (16.9)	0.083	66.3 (13.6)	52.7 (16.6)	0.002	59.1 (13.9)	37.8 (11.0)	0.001

^a^ Independent *t*-test analysis.

## Data Availability

Data Availability Statements are available by requested to corresponding author (orawee.chi@mahidol.ac.th).

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
