# Peer review of "Mobile Application Based Pelvic Floor Muscle Training for Treatment of Stress Urinary Incontinence: An Assessor-Blind, Randomized, Controlled Trial"

_jcm, 2023, doi:10.3390/jcm12227003_

Round 1
Reviewer 1 Report
Comments and Suggestions for Authors
The authors compared the mobile application and traditional PFMT in practical utility. Some concerns should be explained.
1. The main conclusion is that mobile APP increases adherence of patient to exercise, the caption should indicate directly.
2. Compared to pad test, urinary dynamic study is a more appropriate method to confirm SUI, pad test could be used as the evaluation tool referring to treatment outcome.
3. The number of contractions lacks feasibility unless the number of contractions per minute being determined.
4. The indication of asterisk in Table 2.
5. More confounding factors should be incorporated to inclusion or exclusion criteria, such as urinary tract infection, urgency urinary incontinence.
Author Response
Dear Reviewer(s) and Editor.
The article titled " Mobile Application Based Pelvic Floor Muscle Training for Treatment of Stress Urinary Incontinence: An Assessor-blind, Randomized, Controlled Trial” which was submitted to Journal of Clinical Medicine and received the valuable opportunity to be revised by you, has been considered and revised in accordance with your suggestions. Our research team contains six gynecologists who specialize in female pelvic medicine and reconstructive surgery. We intend to make an article that can provide data for better urogynecology healthcare and clinical implications in the future. Thank you for your kindly response and very informative comments to improve the quality of our research. Our research team tries so hard to improve our article to make it the best and get the opportunity to publish in your journal. Moreover, we believe that our article is proper and fits your journal in the special issue, and it will provide interesting data for meta-analysis or cross-country analysis in the future, which can lead to many citations. We are looking forward to getting the good news of our acceptance to publish in your journal.
We would like to inform the editors and all reviewers that we have amended all of the reviewers' suggestions and expanded the article to exceed 4000 words in accordance with journal standards.Furthermore, we had the institute language service rewrite and proofread the English.
Thank you for the constructive comments, suggestions, and critiques. We have responded point-by-point below in RED and addressed them in the manuscript using track changes.
Yours sincerely,
Associate Professor Orawee Chinthakanan
Corresponding author: E-mail: orawee.chi@mahidol.ac.th
Comments and Suggestions for Authors: Reviewer 1
Introduction
- The main conclusion is that mobile APP increases adherence of patient to exercise, the caption should indicate directly.
Response: Thank you so much for the best comments that could improve the quality of our research. We state the main conclusion as your suggestion.
- Compared to pad test, urinary dynamic study is a more appropriate method to confirm SUI, pad test could be used as the evaluation tool referring to treatment outcome.
Response: We appreciate your suggestions. Because UDS is an invasive and costly tool for examination. This is our limitation in using UDS in patient diagnosis.
However, in our design, we declare unambiguously that we will use clinical signs of SUI form patients reported history, Questionnaire to confirm disease, Cough test, and Pad test to confirm SUI in our study participants. Furthermore, we assess the results, which included both objective and subjective evaluations, to ensure that clinical symptoms and progress were correct.
- The number of contractions lacks feasibility unless the number of contractions per minute being determined.
Response: We utilize the patients' reports of SUI episodes to determine one of the measuring subjective outcomes.
- The indication of asterisk in Table 2
Response: We appreciate any comments from reviewers. We edited asterisk in table2 and use asterisk as only the significant results as standard. Thank you for good remind.
- More confounding factors should be incorporated to inclusion or exclusion criteria, such as urinary tract infection, urgency urinary incontinence.
Response: We appreciate feedback from reviewers. We edited the inclusion and exclusion criteria as your suggestion.
Reviewer 2 Report
Comments and Suggestions for Authors
Dear Authors,
We recently had the opportunity to read your manuscript titled “Mobile Application Based Pelvic Floor Muscle Training for Treatment of Stress Urinary Incontinence: An Assessor-blind, Randomized, Controlled Trial”, and we wanted to reach out to you to express our comments about your work.
This randomized controlled trial evaluated a mobile app for pelvic floor muscle training compared to standard instructions for treatment of stress urinary incontinence in women. After 12 weeks, the app group had similar urinary incontinence outcomes but higher adherence to pelvic floor exercises and improved quality of life versus the control group. The app may enhance adherence and symptom improvement in women undergoing conservative treatment for stress urinary incontinence.
Nevertheless, here are some possible comments outlining areas that could improve the quality and readability of the manuscript:
Introduction:
1. Research Research objectives could be stated more clearly and concisely.
2. Additional background information is needed about the prevalence and impact of stress urinary incontinence.
3. Expand the review of prior research on pelvic floor muscle training and mobile apps for incontinence.
Methods:
4. Details about participants recruitment process and criteria should be expanded.
5. Justify the chosen sample size and explain the power calculation.
6. Validate the questionnaires used and reference them in details.
7. The statistical analysis methods need more explanation.
Findings/Results:
8. Organize the Results section more logically flowing from primary to secondary outcomes. It is not very clearly explained.
9. Include basic descriptive statistics for participant groups. Table 1 is not enough.
10. Use more tables/figures to summarize the results data.
11. Report all planned outcome measures.
Discussion:
12. The interpretation of findings is reasonable but could be expanded with reference to other studies.
13. Explain the limitations of the small sample size, and explain why the follow-up should be acknowledged.
14. The Discussion of clinical implications could be strengthened.
Conclusions:
15. Research aims are addressed but the Conclusions could be more clearly tied to key findings and referenced to the aims of the study.
16. The significance and next steps for research could be highlighted more.
While the manuscript is reasonably well-written, there are a number of issues that need to be addressed. Several grammatical errors were noticed including incorrect verb tenses, missing articles, lack of subject-verb agreement, and awkward or unclear phrasing. Some sentences are too long and convoluted. There are also a few spelling and punctuation errors throughout. The writing style could be more concise and precise to improve clarity. Certain sections require reworking for better flow and readability. The overuse of abbreviations and acronyms makes some parts challenging to follow. Overall, in its current state, the manuscript falls below the high standards for English language and grammar . Significant editing is needed to improve clarity, precision, conciseness, and flow of the writing. Tightening up grammar, sentence structure, and phrasing should be prioritized. The authors would benefit from having the entire manuscript thoroughly copyedited by a native English speaker/editor before resubmitting to ensure it meets the expectations of a top-tier journal.
Once again, thank you very much for your work. We´ll be waiting for your answers about our comments.
Kindest regards,
Comments on the Quality of English Language
While the manuscript is reasonably well-written, there are a number of issues that need to be addressed. Several grammatical errors were noticed including incorrect verb tenses, missing articles, lack of subject-verb agreement, and awkward or unclear phrasing. Some sentences are too long and convoluted. There are also a few spelling and punctuation errors throughout. The writing style could be more concise and precise to improve clarity. Certain sections require reworking for better flow and readability. The overuse of abbreviations and acronyms makes some parts challenging to follow. Overall, in its current state, the manuscript falls below the high standards for English language and grammar required . Significant editing is needed to improve clarity, precision, conciseness, and flow of the writing. Tightening up grammar, sentence structure, and phrasing should be prioritized. The authors would benefit from having the entire manuscript thoroughly copyedited by a native English speaker/editor before resubmitting to ensure it meets the expectations of a top-tier journal.
Author Response
POINT-BY-POINT RESPONSES TO THE REVIEWERS’ COMMENTS
13rd October 2023
Dear Reviewer(s) and Editor.
The article titled " Mobile Application Based Pelvic Floor Muscle Training for Treatment of Stress Urinary Incontinence: An Assessor-blind, Randomized, Controlled Trial” which was submitted to Journal of Clinical Medicine and received the valuable opportunity to be revised by you, has been considered and revised in accordance with your suggestions. Our research team contains six gynecologists who specialize in female pelvic medicine and reconstructive surgery. We intend to make an article that can provide data for better urogynecology healthcare and clinical implications in the future. Thank you for your kindly response and very informative comments to improve the quality of our research. Our research team tries so hard to improve our article to make it the best and get the opportunity to publish in your journal. Moreover, we believe that our article is proper and fits your journal in the special issue, and it will provide interesting data for meta-analysis or cross-country analysis in the future, which can lead to many citations. We are looking forward to getting the good news of our acceptance to publish in your journal.
We would like to inform the editors and all reviewers that we have amended all of the reviewers' suggestions and expanded the article to exceed 4000 words in accordance with journal standards.Furthermore, we had the institute language service rewrite and proofread the English.
Thank you for the constructive comments, suggestions, and critiques. We have responded point-by-point below in RED and addressed them in the manuscript using track changes.
Yours sincerely,
Associate Professor Orawee Chinthakanan
Corresponding author: E-mail: orawee.chi@mahidol.ac.th
Comments and Suggestions for Authors: Reviewer 2
Introduction
- Research Research objectives could be stated more clearly and concisely.
Response: Thank you so much for the best comments that could improve the quality of our research. We stated more clearly and concisely objective as your suggestion.
(PAGE 3 , Line 99-103)
- Additional background information is needed about the prevalence and impact of stress urinary incontinence.
Response: Thank you so much for the best comments that could improve the quality of our research. We added more additional background information as your suggestion. (Line 33-36, 40-42)
- Expand the review of prior research on pelvic floor muscle training and mobile apps for incontinence.
Response: Thank you so much for the best comments that could improve the quality of our research. We stated more prior research as your suggestion.
(Page2, Line52-79)
Methods:
- Details about participants recruitment process and criteria should be expanded.
Response: We appreciate your comments. We add more detail of recruitment process as your suggestion.( Page3 Line 112-122)
- Justify the chosen sample size and explain the power calculation.
Response: We appreciate your comments. We add more detail of sample size calculation as your suggestion.( Page4 Line 146-156)
- Validate the questionnaires used and reference them in details.
Response: We appreciate your comments. We add all references of validated questionnaires as your suggestion.
- The statistical analysis methods need more explanation.
Response: We appreciate your comments. We add more explaination of the analysis method as your suggestion.( Page5 Line 198-206)
Response:
Findings/Results:
- Organize the Results section more logically flowing from primary to secondary outcomes. It is not very clearly explained.
Response: We appreciate your comments. This is very important to improve our quality of manuscript. We revised the flowing of results with clearly explaination as your suggestion. (PAGE6-7, Line 238-262)
- Include basic descriptive statistics for participant groups. Table 1 is not enough.
Response: We appreciate your comments. This is very important to improve our quality of manuscript. We revised the flowing of results with clearly explaination as your suggestion. (PAGE6, Line 220-236)
- Use more tables/figures to summarize the results data.
Response: We appreciate your comments. We explain and adjust data to more easy for understanding as your suggestion.
- Report all planned outcome measures.
Response: We appreciate your comments. This is very important to improve our quality of manuscript. (Line 182-188)
Discussion:
- The interpretation of findings is reasonable but could be expanded with reference to other studies.
Response: We respect the reviewer's opinion. We expanded the reference as your suggestion)
- Explain the limitations of the small sample size and explain why the follow-up should be acknowledged.
Response: We respect the reviewer's opinion. We explained as your suggestion and add more specific details with reference. Thank you so much for this comments.
- The Discussion of clinical implications could be strengthened.
Response: We respect the reviewer's opinion. We revised and added more specific details with wide range of clinical implications.
Response:
Conclusions:
- Research aims are addressed but the Conclusions could be more clearly tied to key findings and referenced to the aims of the study.
Response: We respect the reviewer's opinion. We revised and added more specific details as your suggestion.
- The significance and next steps for research could be highlighted more.
Response: We respect the reviewer's opinion. We revised and added more specific details with wide range of next steps for our research.
Reviewer 3 Report
Comments and Suggestions for Authors
The study is well designed that tried to find answers for the use of pelvic floor muscle training with mobile phone on stress urinary incontinence.
It is a prospective randomized study. the inclusion and exclusion criteria are well defined. sample size analysis is done. primary and secondary outcomes are defined.
ı would suggest one minor suggestion as
1) please mention some other treatment option for pelvic floor muscle training. for example with electrical stimulation. please refere this randomized study:DOI: 10.5505/ejm.2020.87609
Author Response
POINT-BY-POINT RESPONSES TO THE REVIEWERS’ COMMENTS
13rd October 2023
Dear Reviewer(s) and Editor.
The article titled " Mobile Application Based Pelvic Floor Muscle Training for Treatment of Stress Urinary Incontinence: An Assessor-blind, Randomized, Controlled Trial” which was submitted to Journal of Clinical Medicine and received the valuable opportunity to be revised by you, has been considered and revised in accordance with your suggestions. Our research team contains six gynecologists who specialize in female pelvic medicine and reconstructive surgery. We intend to make an article that can provide data for better urogynecology healthcare and clinical implications in the future. Thank you for your kindly response and very informative comments to improve the quality of our research. Our research team tries so hard to improve our article to make it the best and get the opportunity to publish in your journal. Moreover, we believe that our article is proper and fits your journal in the special issue, and it will provide interesting data for meta-analysis or cross-country analysis in the future, which can lead to many citations. We are looking forward to getting the good news of our acceptance to publish in your journal.
We would like to inform the editors and all reviewers that we have amended all of the reviewers' suggestions and expanded the article to exceed 4000 words in accordance with journal standards.Furthermore, we had the institute language service rewrite and proofread the English.
Thank you for the constructive comments, suggestions, and critiques. We have responded point-by-point below in RED and addressed them in the manuscript using track changes.
Yours sincerely,
Associate Professor Orawee Chinthakanan
Corresponding author: E-mail: orawee.chi@mahidol.ac.th
Comments and Suggestions for Authors: Reviewer 3
The study is well designed that tried to find answers for the use of pelvic floor muscle training with mobile phone on stress urinary incontinence.
It is a prospective randomized study. the inclusion and exclusion criteria are well defined. sample size analysis is done. primary and secondary outcomes are defined.
ı would suggest one minor suggestion as
1) please mention some other treatment option for pelvic floor muscle training. for example with electrical stimulation. please refere this randomized study:DOI: 10.5505/ejm.2020.87609
Response: We appreciate the reviewer's comments. We have revised the manuscript as all suggestions above. However, the reference above has no relate with our trial.
Thank you for your good comments and appreciate our research.
Round 2
Reviewer 1 Report
Comments and Suggestions for Authors
All the concerns are explained, and I have no other question.
Author Response
Dear Reviewer(s) and Editor.
The article titled " Mobile Application Based Pelvic Floor Muscle Training for Treatment of Stress Urinary Incontinence: An Assessor-blind, Randomized, Controlled Trial” which was submitted to Journal of Clinical Medicine and received the valuable opportunity to be revised by you, has been considered and revised in accordance with your suggestions. Our research team contains six gynecologists who specialize in female pelvic medicine and reconstructive surgery. We intend to make an article that can provide data for better urogynecology healthcare and clinical implications in the future. Thank you for your kindly response and very informative comments to improve the quality of our research. Our research team tries so hard to improve our article to make it the best and get the opportunity to publish in your journal. Moreover, we believe that our article is proper and fits your journal in the special issue, and it will provide interesting data for meta-analysis or cross-country analysis in the future, which can lead to many citations. We are looking forward to getting the good news of our acceptance to publish in your journal.
We would like to inform the editors and all reviewers that we have amended all of the reviewers' suggestions.
Thank you for the constructive comments, suggestions, and critiques. We have responded point-by-point below in RED and addressed them in the manuscript using track changes.
Yours sincerely,
Associate Professor Orawee Chinthakanan
Corresponding author: E-mail: orawee.chi@mahidol.ac.th
Comments and Suggestions for Authors: Reviewer 1
All the concerns are explained, and I have no other question.
Response: Thank you for your consideration.
Reviewer 2 Report
Comments and Suggestions for Authors
Dear Authors,
Thank you for the opportunity to review the revised version of your manuscript "Mobile Application Based Pelvic Floor Muscle Training for Treatment of Stress Urinary Incontinence: An Assessor-blind, Randomized, Controlled Trial". I appreciate you taking the time to address the comments from the previous review. However, upon examining the manuscript, I believe there are still some significant issues that need to be improved.
While the additional details you have provided in the Methods section are helpful, the Introduction requires expansion of the background information on stress urinary incontinence prevalence, impact, and prior research on pelvic floor muscle training and mobile applications. The objectives and purpose of the study need to be stated more precisely and the last sentence of the aims of the study does not fit correctly because in the aims you are talking about the purpose of the study and that last sentence is written in the past tense. In the Methods, further information is needed on the recruitment process such as inclusion/exclusion criteria, and specifics of the statistical analyses performed. As you may see, in the Methods section, only the Wilcoxon test is mentioned (lines 204-205) but in Table 1 and Table 2 of the Results section, the Independent t-test, Man-Whitney U test and the Wilcoxon signed rank test are performed in order to compare App Group and Control Group. This is something quite relevant that must be modified. The sample size justification and power calculation also require better explanation.
The presentation of results appears disorganized and does not flow logically from primary to secondary outcomes. Correcting the tables or figures would help summarize the data and findings clearly. It seems that some planned outcome measures are not reported. The Discussion lacks sufficient comparison to findings in the literature and does not expand upon the limitations related to sample size and short follow-up. Additionally, the clinical implications could be explored further. The Conclusion do not adequately tie back to the original aims of the study, and the significance of the research is not highlighted. With this comment I would like to detail the following:
The three objectives of this study were to evaluate:
1. the effects of using the mobile pelvic floor muscle exercise app on the cure rate of stress urinary incontinence (SUI).
2. the frequency of SUI episodes.
3. patients' adherence to pelvic floor muscle exercises at home over a 12-week period.
And in the Conclusion it only refers to the third objective (improving adherence to PFMT) without mentioning the first two aims. But, in addition, it states that "The pelvic floor muscle exercise application […] improves symptoms in women with stress urinary incontinence", which is something that does not appear in the objectives of the study and it is not clear if the conducted research was measuring this research question. It is mandatory to correct these aspects both in the aims and in the Conclusions.
I am concerned that issues with grammar, sentence structure, clarity, conciseness and flow persist throughout the manuscript. Moderate editing appears necessary to improve the quality of writing and meet the standards of the journal.
While I appreciate your efforts to revise the manuscript, the major issues identified in the initial review do not seem to be adequately addressed yet. Significant work remains to be done to improve the Introduction, Methods, Results, Discussion and Conclusions sections as outlined above. I would be happy to reconsider a revised version after you have thoroughly addressed these concerns. Please feel free to contact me if you would like to discuss specific ways to improve the paper. I hope these comments are helpful as you continue to refine this research study into a strong journal submission.
Once again, thank you very much for your work. We´ll be waiting for your answers about our comments.
Kindest regards,
Comments on the Quality of English Language
I am concerned that issues with grammar, sentence structure, clarity, conciseness and flow persist throughout the manuscript. Moderate editing appears necessary to improve the quality of writing and meet the standards of the journal.
Author Response
Dear Reviewer(s) and Editor.
The article titled " Mobile Application Based Pelvic Floor Muscle Training for Treatment of Stress Urinary Incontinence: An Assessor-blind, Randomized, Controlled Trial” which was submitted to Journal of Clinical Medicine and received the valuable opportunity to be revised by you, has been considered and revised in accordance with your suggestions. Our research team contains six gynecologists who specialize in female pelvic medicine and reconstructive surgery. We intend to make an article that can provide data for better urogynecology healthcare and clinical implications in the future. Thank you for your kindly response and very informative comments to improve the quality of our research. Our research team tries so hard to improve our article to make it the best and get the opportunity to publish in your journal. Moreover, we believe that our article is proper and fits your journal in the special issue, and it will provide interesting data for meta-analysis or cross-country analysis in the future, which can lead to many citations. We are looking forward to getting the good news of our acceptance to publish in your journal.
We would like to inform the editors and all reviewers that we have amended all of the reviewers' suggestions.
Thank you for the constructive comments, suggestions, and critiques. We have responded point-by-point below in RED and addressed them in the manuscript using track changes.
Yours sincerely,
Associate Professor Orawee Chinthakanan
Corresponding author: E-mail: orawee.chi@mahidol.ac.th
Comments and Suggestions for Authors: Reviewer 1
All the concerns are explained, and I have no other question.
Response: Thank you for your consideration.
Comments and Suggestions for Authors: Reviewer 2
While the additional details you have provided in the Methods section are helpful, the Introduction requires expansion of the background information on stress urinary incontinence prevalence, impact, and prior research on pelvic floor muscle training and mobile applications.
Response:
- The review of prevalence was added more in the paragraph 1 of introduction.
- The expansion of the background information on SUI impact was added as your suggestion.
- The prior research which reflects how important to study in this topic was added in the introduction.
The objectives and purpose of the study need to be stated more precisely and the last sentence of the aims of the study does not fit correctly because in the aims you are talking about the purpose of the study and that last sentence is written in the past tense.
Response: Thank you for the suggestions. We appreciate this suggestion to improve our manuscript. We revised the sentence of objectives and purpose of the study for the precision and correct as your suggestion.
In the Methods, further information is needed on the recruitment process such as inclusion/exclusion criteria, and specifics of the statistical analyses performed.
As you may see, in the Methods section, only the Wilcoxon test is mentioned (lines 204-205) but in Table 1 and Table 2 of the Results section, the Independent t-test, Man-Whitney U test and the Wilcoxon signed rank test are performed in order to compare App Group and Control Group. This is something quite relevant that must be modified.
The sample size justification and power calculation also require better explanation.
Response: Thank you for suggestion. We agree with reviewer and revised as your suggestion
The presentation of results appears disorganized and does not flow logically from primary to secondary outcomes. Correcting the tables or figures would help summarize the data and findings clearly. It seems that some planned outcome measures are not reported.
Response: Thank you for your best comments to improve our manuscript.
We really agree with you and try to improve this section since last revision
Now we re-order to be the same order since the material and methods section until the results section with confirm the relevant and orderly.
For the presentation of results
- First paragraph: Description the result of Patients’ recruitment and study flow (with correlated with Figure1 CONSORT Flowchart)
- Second paragraph: Description of patients baseline characteristic and pre-treatment severity of SUI (with correlated with Table1) – We add more description into the details as you suggestion from the first time of revision.
- Third paragraph: Description of Primary outcome which is objective measurement (one-hour pad test) (with correlated with Table2 in Line 1 and 2 of the table)
In the third paragraph for the results is correlate and cover all the outcome measurements that state in “Main outcome measurement” in materials and methods section
- Fourth paragraph: Description of Secondary outcomes including SUI episode, PGI-I, ICIQ-UI SF. (with correlated with Table2)
In the fourth paragraph in the results correlate with state in section “secondary outcomes.” We revised section “the secondary outcomes” in methods relevant to the results.
- Fifth paragraph: Description of adherence in PFMT which is correlate to Figure3 and Table 3
The Discussion lacks sufficient comparison to findings in the literature and does not expand upon the limitations related to sample size and short follow-up. Additionally, the clinical implications could be explored further.
Response: Thank you for your suggestion.
- We added comparison of findings in various aspect as your suggestion
- We added more details in discussion and expand the limitations related to sample size and loss-follow up.
- We added more clinical implications and further research in discussion section
The Conclusion do not adequately tie back to the original aims of the study, and the significance of the research is not highlighted. With this comment I would like to detail the following:
The three objectives of this study were to evaluate:
- the effects of using the mobile pelvic floor muscle exercise app on the cure rate of stress urinary incontinence (SUI).
- the frequency of SUI episodes.
- patients' adherence to pelvic floor muscle exercises at home over a 12-week period.
And in the Conclusion it only refers to the third objective (improving adherence to PFMT) without mentioning the first two aims. But, in addition, it states that "The pelvic floor muscle exercise application […] improves symptoms in women with stress urinary incontinence", which is something that does not appear in the objectives of the study and it is not clear if the conducted research was measuring this research question. It is mandatory to correct these aspects both in the aims and in the Conclusions.
Response: Thank you for your comments. We agree with you. We revised the conclusion as you suggestion.
Round 3
Reviewer 2 Report
Comments and Suggestions for Authors
Dear Authors,
Thank you for the opportunity to review the third revised version of your manuscript "Mobile Application Based Pelvic Floor Muscle Training for Treatment of Stress Urinary Incontinence: An Assessor-blind, Randomized, Controlled Trial". I appreciate you taking the time to address the comments from the previous review. Nevertheless, after reading this new version of the manuscript, I would recommend you to follow these comments:
- The modifications in the manuscript don´t appear easily due to the fact that different colors are used in the text, with yellow highlights and normal writing. Please, prepare the document for ease the reading. I would recommend you to mention the lines numbers of each change as I did in some of my previous comments.
- There are some comments in the change tracker written in Thai language, which is something quite unusual. I don´t know if I have to translate them for reading them or just forget about them. If this version of the manuscript is for reviewers, please adapt it for reviewers.
- It is really difficult to read this version because of many sentences which make no sense, such as: “adherence were was performed by independent t-test”, “were expressed as quantity frequency and percentage” or “Chi-square test was used to compare the groups in realtion to the qualitative variables. compared using chi-square test”. Several spelling and grammar mistakes can also be found in the text which also makes understanding even harder. Moderate editing should be done by a native speaker.
- Some of our previous comments are still waiting for being addressed and, specially, those related to the aims and conclusions of the research. I still find that they don´t match each other.
While I appreciate your efforts to revise the manuscript, these comments of the previous review do not seem to be adequately addressed yet.
As said before, I would be happy to review a new version of your manuscript after you have modified it accordingly.
Thank you very much for your work.
Kindest regards,
Comments on the Quality of English Language
Moderate editing should be done by a native speaker.
Author Response
Dear Reviewers
We apologize for the technical error that occurred with the files.We carefully revised the manuscript in accordance with all of your suggestions.
Please reconsider the attached manuscript file and response file.
We hope that this final version will satisfy you and give us the chance to publish in this journal.
Authors Team
Round 4
Reviewer 2 Report
Comments and Suggestions for Authors
Dear Authors,
Thank you for the opportunity to review the fourth revised version of your manuscript "Mobile Application Based Pelvic Floor Muscle Training for Treatment of Stress Urinary Incontinence: An Assessor-blind, Randomized, Controlled Trial". At this stage, I only have a few minor recommendations to further improve the manuscript:
· The sample size justification and power calculation need explanation.
· In the Results, double check the logical flow and organization of outcomes from primary to secondary.
· The Discussion lacks comparison to findings in other literature and does not expand on limitations like small sample size and short follow-up.
· Do a final check for grammar, spelling, and sentence structure throughout.
With just these minor edits, I believe your manuscript will increase its quality.
Thank you very much again for your work and patience throughout the review process.
Kindest regards,
Comments on the Quality of English Language
Do a final check for grammar, spelling, and sentence structure throughout.
Author Response
Thank you for your suggestion
We hope that this version will make you appreaciate.